# EMERGENCE OF LINGUISTIC COMMUNICATION FROM REFERENTIAL GAMES WITH SYMBOLIC AND PIXEL INPUT

**Angeliki Lazaridou**\*, **Karl Moritz Hermann, Karl Tuyls, Stephen Clark**
DeepMind,
London, UK

## ABSTRACT

The ability of algorithms to evolve or learn (compositional) communication protocols has traditionally been studied in the language evolution literature through the use of emergent communication tasks. Here we scale up this research by using contemporary deep learning methods and by training reinforcement-learning neural network agents on referential communication games. We extend previous work, in which agents were trained in symbolic environments, by developing agents which are able to learn from raw pixel data, a more challenging and realistic input representation. We find that the degree of structure found in the input data affects the nature of the emerged protocols, and thereby corroborate the hypothesis that structured compositional language is most likely to emerge when agents perceive the world as being structured.

## 1 INTRODUCTION

The study of emergent communication is important for two related problems in language development, both human and artificial: language evolution, the development of communication protocols from scratch (Nowak & Krakauer, 1999); and language acquisition, the ability of an embodied agent to learn an existing language. In this paper we focus on the problem of how environmental or pre-linguistic conditions affect the nature of the communication protocol that an agent learns. The increasing realism and complexity of environments being used for grounded language learning (Brockman et al., 2016; Hermann et al., 2017) present an opportunity to analyse these effects in detail.

In line with previous work on emergent communication, we are strongly motivated by the view that language derives meaning from its use (Wittgenstein, 1953; Wagner et al., 2003). This perspective especially motivates the study of language emergence in cases where co-operative agents try to achieve shared goals in game scenarios (Steels, 2003; Brighton & Kirby, 2006; Mordatch & Abbeel, 2017), and is related to the study of multi-agent and self-play methods that have found great success in other areas of machine learning (Bansal et al., 2017; Silver et al., 2017). Here we focus on simple referential games, in which one agent must communicate to another a target object in the agent's environment.

One of the most important properties of natural language is compositionality. Smaller building blocks (e.g. words, morphemes) are used to generate unbounded numbers of more complex forms (e.g. sentences, multi-word expressions), with the meaning of the larger form being determined by the meanings of its parts and how they are put together (Frege, 1892). Compositionality is an advantage in any communication protocol as it allows in principle infinite expression through a finite dictionary and a finite set of combination rules. In emergent communication research, previous work has shown that agents can produce (somewhat) compositional protocols when engaging in language games (Steels, 2003). However, the computational agents were typically situated in artificial worlds containing just a handful of objects, represented as disentangled, structured, and sometimes even atomic symbols, e.g. attribute-based or one-hot vectors (Batali, 1998; Brighton, 2002; Franke, 2015; Andreas & Klein, 2017; Mordatch & Abbeel, 2017). However, humans receive raw sensorimotor

---

\*Corresponding author: `angeliki@google.com`

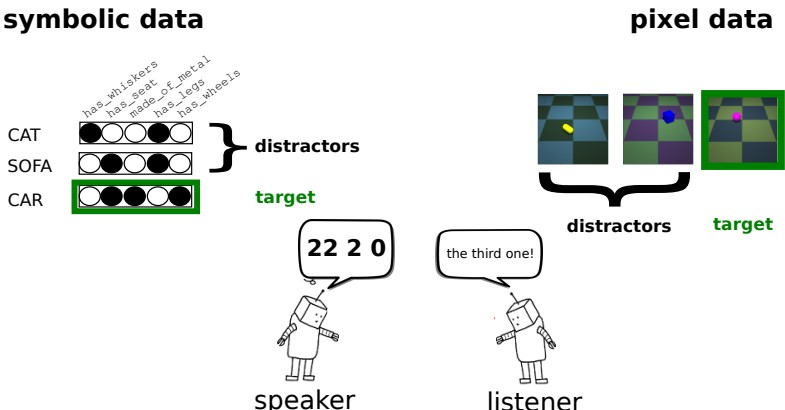

Figure 1: High-level overview of the referential game.

rather than symbolic input, and little work to date has tested whether these findings carry over when agents are situated in less idealized worlds that bear more similarity to the kind of entangled and noisy environments to which humans are typically exposed.[1]

In this work, in the context of referential communication games (see Figure 1), we contrast the results of two studies that lie at the extremes of how much structure is provided by the environment. The first study (Section 3) focuses on symbolic representations, where objects are represented as bags-of-attributes; this representation is inherently disentangled since dimensions encode individual properties. The second study (Section 4) considers raw perceptual input, hence data that more closely resembles what humans are exposed to. Clearly, the latter is a more challenging and realistic scenario as the computational agents are operating on entangled inputs with no pre-coded semantics. Crucially, both studies use the same referential game setup, the same learning procedure (policy learning methods) and the same neural network agent architectures.

We show that reinforcement learning agents can successfully communicate, not only when presented with symbolic and highly structured input data, but (and more importantly) even when presented with raw pixel input. This result opens up the possibility of more realistic simulations of language emergence. We successfully use the learning signal from the referential game to train agents end-to-end, including cases where the agents need to perform visual processing of images with a convolutional neural network. However, we find that the agents struggle to produce structured messages when presented with entangled input data (Bengio et al., 2013) due to the difficulty of uncovering the true factors of variation, corroborating the hypothesis of Smith et al. (2003) that structured (compositional) language is most likely to emerge when agents perceive the world as structured.

## 2 REFERENTIAL GAMES AS MULTI-AGENT CO-OPERATIVE REINFORCEMENT LEARNING

The referential game is implemented as an instance of multi-agent co-operative reinforcement learning, in which two agents take discrete actions in their environment in order to maximize a shared reward.

### 2.1 GAME AND TERMINOLOGY

The referential game is a variant of the Lewis signaling game (Lewis, 1969), which has been extensively used in linguistic and cognitive studies in the context of language evolution (e.g., Briscoe, 2002; Cangelosi & Parisi, 2002; Steels & Loetzsch, 2012; Spike et al., 2016; Lazaridou et al., 2017).

---

[1]An exception is the recent work by Havrylov & Titov (2017), Evtimova et al. (2017) and Lazaridou et al. (2017). However, the authors consider pre-trained visual input which already encodes semantics of the world in the form of the presence of objects and their properties.

Figure 1 provides a schematic description of our setup. First, a *speaker* is presented with a target object (highlighted as *CAR* in the symbolic example on the left, and highlighted as the far right image in the pixel example on the right). Then, by making use of an *alphabet* consisting of primitive discrete *symbols* ("22", "10", "0","2"), the speaker constructs a *message* describing that object ("22 2 0"). We will refer to the set of all distinct messages generated by the speaker as their *lexicon* or *protocol*. Finally, the *listener* is presented with the target and a set of distractor objects, and—by making use of the speaker's message—has to identify the target object from the set of candidate objects. *Communicative success* is defined as the correct identification of the target by the listening agent.

Formally, the attribute-based object vectors (disentangled) or the pixel-based images (entangled) are the set of *pre-linguistic* items $W = \{o_1, \ldots, o_N\}$. From this set we draw a target $t \in W$ and subsequently $K - 1$ distractors $D = \{d_1, \ldots, d_{K-1}\} \subset W$ s.t. $\forall j\ t \neq d_j$. The speaker has only access to the target $t$, while the listener receives candidate set $C = t \cup D$, not knowing which of the elements in $C$ is target $t$.

## 2.2 Agents

The speaker encodes $t$ into a dense representation $u$ using an encoder $f^S(t, \theta_f^S)$. The function of this encoder depends on the type of pre-linguistic data used and is discussed separately for each study. Given an alphabet $A$ of discrete unit symbols (akin to words) and $u$, the speaker next generates a discrete, variable-length, bounded message $\mathbf{m}$ by sampling symbols from a recurrent policy $\pi^S$ defined by a decoder $g^S(u, \theta_g^S)$. The sequence generation is terminated either by the production of a stop symbol or when the maximum length $L$ has been reached. We implement the decoder as a single-layer LSTM (Hochreiter & Schmidhuber, 1997). Note that the symbols in the agents' alphabet $A$ have no *a priori* meaning; rather, these symbols get grounded during the game.

The listening agent uses a similar encoder to the speaker but has independent network weights ($\theta_f^L$). Applying this encoder to all candidate objects results in a set $U = \{f^L(c, \theta_f^L) \mid c \in C\}$. For encoding the message $\mathbf{m}$, we use a single-layer LSTM, denoted $h^L$, which produces an encoding $z$: $z = h^L(\mathbf{m}, \theta_h^L)$.

Given encoded message $z$ and candidates $U$, the listener predicts a target object $t' \in C$ following a policy $\pi^L$ implemented using a non-parametric pointing module; this module samples the predicted object from a Gibbs distribution computed via the dot product between vector $z$ and all encoded candidates $u \in U$. See Appendix B for information regarding the agents' architecture.

At inference time, we replace the stochastic sampling of the speaker's message and the listener's stochastic pointing module with deterministic processes. For the pointing module, the object with the highest probability is chosen. For the speaker's message, this is generated in a greedy fashion by selecting the highest-probability symbol at each step.

## 2.3 Learning

All weights of the speaker and listener agents, $\theta = \{\theta_f^S, \theta_g^S, \theta_f^L, \theta_h^L\}$, are jointly optimized while playing the game. We emphasize that no weights are shared between the speaker and the listener, and the only supervision used is communicative success, i.e. whether the listener identified the correct target. The objective function that the two agents maximize for one training instance is:

$$R(t') \left( \sum_{l=1}^{L} \log p(m_t^l | m_t^{<l}, u) + \log p(u_{t'} | z, U) \right)$$

where $R$ is the reward function returning 1 if $t = t'$ (if the listener pointed to the correct target) and 0 otherwise. To maintain exploration in the speaker's policy $\pi^S$ of generating a message, and the listener's policy $\pi^L$ of pointing to the target, we add to the loss an entropy regularization term (Mnih et al., 2016). The parameters are estimated using the REINFORCE update rule (Williams, 1992). See Appendix B for more details regarding the learning.

| max length | alphabet size | lexicon size | training accuracy | topographic $\rho$ |
|---|---|---|---|---|
| 2 | 10 | 31 | 92.0% | 0.13 |
| 5 | 17 | 293 | 98.2% | 0.16 |
| 10 | 40 | 355 | 98.5% | 0.26 |

Table 1: Commumicative success (**training accuracy** in percentage) with varying maximum message length. **alphabet size** denotes the effective size of the symbol set used from a maximum of 100. **lexicon size** is the effective number of unique messages used. **topographic** $\rho$ reports the structural similarity in terms of Spearman $\rho$ correlation between the message and the object vector space. All Spearman $\rho$ correlations throughout the paper are significant with $p < 0.01$.

## 3  STUDY 1: REFERENTIAL GAME WITH SYMBOLIC DATA

We first present experiments where agents are learning to communicate when presented with structured and disentangled input. We use the Visual Attributes for Concepts Dataset (VisA) of Silberer et al. (2013), which contains human-generated per-concept attribute annotations for 500 concrete concepts (e.g., *cat*, *sofa*, *car*) spanning across different categories (e.g., *mammals*, *furniture*, *vehicles*), annotated with 636 general attributes (e.g., *has_tail*, *is_black*, *has_wheels*). We disregarded homonym concepts (e.g., *bat*), thus reducing our working set of concepts to 463 and the number of attributes to 573 (after eliminating any attribute that did not occur with the working concepts). On average, each concept has 11 attributes. All pre-linguistic objects are represented in terms of binary vectors $o \in \{0, 1\}^{573}$. Note that these representations do carry some inherent structure; the dimensions in the object vectors are disentangled and so each object can be seen as a conjunction of properties. Speaker and listener convert the pre-linguistic representations to dense representations $u$ by using a single-layer MLP with a sigmoid activation function.

In all experiments, we set the number of candidate objects $K$ to five, meaning there were four wrong choices per correct one (resulting in a 20% random baseline). Inspired by Kottur et al. (2017), who show that non-compositional language emerges in the case of overcomplete alphabets, we set the size of alphabet $A$ to 100 symbols, which is smaller than the size of the set of objects (463).

### 3.1  AGENT PERFORMANCE AND AMBIGUITY

We first report model performance on the training data, comparing different settings for the maximal allowed message length (2, 5 or 10 symbols). Results are presented in Table 1 (ignore the last row **topographic** $\rho$ which will be explained in later sections).

In the case of the shortest message settings (maximum length 2), our trained agents on average only develop a protocol of 31 unique messages used to describe 363 training concepts (leaving aside 100 for testing). This indicates high levels of ambiguity, with each message being used to denote 11 concepts on average. Interestingly, recent findings suggest that ambiguity is a design feature of language that prevents the inefficient use of redundant codes, since some of the message content can be extracted from context: "the most efficient communication system will not convey information already provided by the context" (Piantadosi et al., 2012). In our case, we do no explicitly encode any bias towards ambiguity. We hypothesize that ambiguity arises due to the difficult exploration problem that agents are faced with, in combination with the fact that ambiguous protocols present a good local optimum that is over-represented in the hypothesis search space. As a result, in the absence of environmental pressures (e.g., a high number of carefully constructed distractors) a suboptimal policy can still achieve a reasonably high accuracy (92%), making it even harder during training to escape from such a solution.

In classic signaling games, this polysemy phenomenon manifests itself as different states receiving the same signal and is termed *partial pooling equilibrium* (Skyrms, 2010). Perhaps rather counterintuitively, Skyrms (p.131) suggests that a way to obtain communication protocols that are robust to this type of local communication minima is to allow the invention of new signals, essentially increasing the search space of signals. Motivated by this suggestion, we play variants of the game in which we allow the agents to produce messages of greater maximum length (5 and 10), which leads to improved communicative success (98.2% and 98.5% respectively). We observe that the number

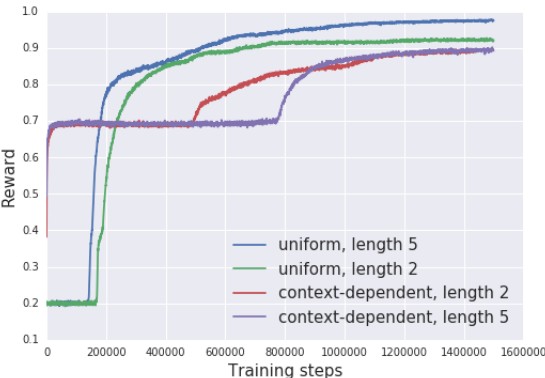

Figure 2: Training curves of different experimental setups with uniform and context-dependent target selection.

of messages in the protocol increases from 31 to 293 and 355, respectively, reducing the average number of concepts a message can denote from 11 concepts to (approximately) 1 concept.

## 3.2 REALISTIC CONTEXT DISTRIBUTION

In the real world, when speakers refer to cats, listeners would likely be in a situation where they had to discriminate a cat in the context of a couch or a dog, rather than in the context of a mirror or a cow.[2] Simply put, objects in the world do not appear in random contexts, but rather there is regularity in the distribution of situational and visual co-occurrences. This property of the world is typically not captured in referential games studied in the language emergence literature, with distractors usually drawn from a uniform distribution.

We address this issue and design an additional experiment with distractors sampled from a target-specific context distribution reflecting normalized object co-occurrence statistics. Co-occurrence data is extracted from the MSCOCO caption dataset (Lin et al., 2014). This leads to more plausible distractor sets with, for instance, the target *goat* more likely being mixed with *sheep* and *cow* as distractors rather than *bike* or *eggplant*.

We find that the distractor selection process (uniform vs context-dependent) affects the language learning dynamics; see Figure 2 for training curves for different experimental configurations. While the non-uniform distractor sampling of the context-dependent setting can be exploited to learn a degenerate strategy —giving up to 40% communicative success shortly after the start of training— subsequently learning under this scenario takes longer. This effect is likely a combination of the local minimum achieved by the degenerate strategy of picking a target at random from only the topically relevant set of distractors, which initially makes the problem easier; however, the fact that the co-occurrence statistics tend to align with the feature vectors, means that similar objects are more likely to appear as distractors and hence the overall game becomes more difficult.

We now consider the question of how objects denoted by the same (ambiguous) message are related. When the context is drawn uniformly, object similarity is a predictor of object *confusability*, as similar objects tend to be mapped onto the same message (0.26 and 0.43 median pairwise cosine similarities of objects that received the same message as computed on the VisA space, for maximum message length 2 and 5, respectively). In the non-uniform case, we observe object confusability to be less influenced by object similarity (0.15 and 0.17 median pairwise cosine similarities of objects that received the same message, for maximum message length 2 and 5, respectively), but rather driven by the visual context co-occurrences. Simply put, in the non-uniform case confusability is less influenced by similarity since the agents must learn to distinguish between objects that naturally co-occur (e.g. *sheep* and *goat*). Thus, the choice of distractors, an experimental design decision that in existing language emergence literature has been neglected, has an effect on the organization (and

---

[2]This example reflects real co-occurrence statistics from caption data.

| Data | length 2 lexicon size | acc. | length 5 lexicon size | acc. | length 10 lexicon size | acc. |
|---|---|---|---|---|---|---|
| training data | 31 | 92.0 | 293 | 98.2 | 355 | 98.5 |
| test data | 1 | 74.2 | 70 | 76.8 | 98 | 81.6 |
| unigram chimera | 5 | 39.3 | 88 | 40.5 | 99 | 47.0 |
| uniform chimera | 3 | 31.2 | 87 | 32.2 | 100 | 42.6 |

Table 2: Communicative success (**acc** in percentage) of agents evaluated on training (first row) and novel (last three rows) data. **lexicon size** column reports the percentage of novel messages (i.e., messages that were not used during the training).

potentially the naturalness) of the emerged language, for example as reflected in the semantics of ambiguous or homonym words in the language.

### 3.3 STRUCTURAL PROPERTIES OF EMERGED PROTOCOLS

Quantifying the degree of compositionality and structure found in the emerged language is a challenging task; to the best of our knowledge, there is no formal mathematical definition of compositionality that would allow for a definitive quantitative measure. Thus, research on this topic usually relies on defining necessary requirements that any language claiming to be compositional should adhere to, such as the ability to generalize to novel situations (Batali, 1998; Franke, 2015; Kottur et al., 2017). We adopt a similar strategy by measuring the extent to which an emerged language is able to generalize to novel objects (Section 3.3.1). Moreover, we also report quantitative results (Section 3.3.2) using a measure of message structure proposed in the language evolution literature (Brighton & Kirby, 2006; Carr et al., 2017).

#### 3.3.1 GENERALIZATION TO NOVEL OBJECTS

We perform experiments where trained agents from Section 3.1 are exposed to different types of unseen objects, each of them differing to the degree to which the unseen objects resemble the objects found in the training data. In the **test** scenario, objects come from the same data distribution as the training data, but were not presented to the agents during training (e.g., a mouse); in the **unigram chimeras** scenario, the novel objects are constructed by sampling properties from a property-based distribution inferred from the training data, thus breaking any feature correlation (e.g., a mouse-like animal with wheels); in the **uniform chimeras** scenario, the novel objects are constructed by uniformly sampling properties (e.g., a square red furry metallic object).

Table 2 reports the communicative success. While there is a drop in performance for unseen objects, agents are performing above random chance (20%). The emerged language is indeed able to generalize to unseen objects; however, the degree of generalization is a function of the similarity between the training and unseen objects, thus resulting in the **uniform chimeras** setting obtaining the lowest performance.

Moreover, we observe examples of *productivity*, a key feature of compositionality. At test time, speakers are able to concoct novel messages on-the-fly (i.e., messages that are not part of their lexicon induced during training) to describe unseen objects. See the last three rows of Table 2, and the **lexicon size** column, for the percentage of novel messages. Even though listeners were not trained to associate novel messages with novel objects, they are still able comprehend such messages and correctly identify the target object. In the **test data** and **length 10** cases, novel messages account for almost all of the generated messages, but with performance at 81.6%, providing evidence of the structure found in the messages.

#### 3.3.2 TOPOGRAPHIC SIMILARITY

Given a set of objects, their meanings and the associated signals, Brighton & Kirby (2006) define *topographic similarity* to be the correlation of the distances between all the possible pairs of meanings and the corresponding pairs of signals. Figure 3 shows mappings between states and signals for examples of holistic (b) and compositional (c,d) languages, with the topographic similarity of

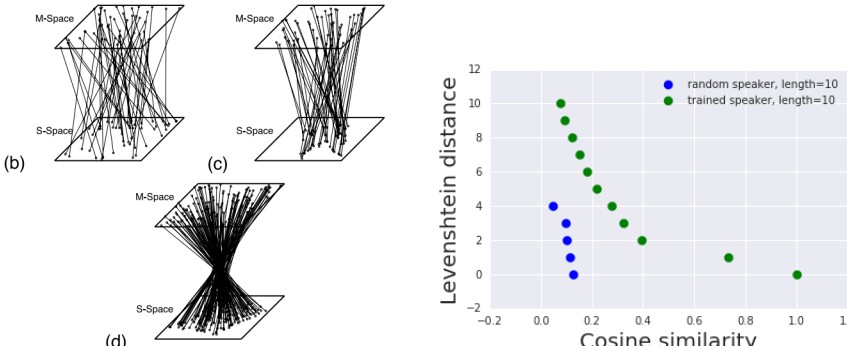

Figure 3: **left**: Three languages with different properties, taken from Brighton & Kirby (2006). The mapping between states and signals shown in (b) is random; there is no relationship between points in the meaning and signal space. In (c) and (d), similar meanings map to similar signals, i.e., there is a topographic relation between meanings and signals. **right**: Relation between objects' cosine similarity and their message Levenshtein distance for trained and random agents.

compositional languages being higher than that of holistic. The intuition behind this measure is that semantically similar objects should have similar messages.

To compute this measure, we first compute two lists of numbers: (i) the Levenshtein distances between all pairs of objects' messages; and (ii) the cosine similarity between all pairs of objects' VisA vectors. Given these two lists, the topographic similarity is defined as their negative Spearman $\rho$ correlation (since we are correlating distances with similarities, negative values of correlation indicate topographic similarity of the two spaces). Intuitively, if similar objects share much of the message structure (e.g., common prefixes or suffixes), and dissimilar objects have little common structure in their respective messages, then the topographic similarity should be high, the highest possible value being 1.

Results presented back in Table 1, in the **topographic** $\rho$ column, show that topographic similarity is positive in all experimental setups, indicating that similar objects receive similar messages ($p < 0.01$, permutation test). A qualitative analysis of the messages generated in the **length 10** and **training data** cases showed that, for example, 32% of the *mammal* objects had as a message prefix the bigram '95#10'; 36% of *vehicle* objects had '68#95'; and 11% of *tool* objects had '0#61', suggesting that these prefix bigrams encode category-specific information.

Next, for each object pair, we calculate their Levenshtein message distance and respective cosine similarity, and plot in Figure 3 (**right**), for each distance, the average cosine similarities of the pairs with that distance (this is done for the **length 10** and **training data** experiment). We observe that there is a clear relation between message similarity and meaning similarity (as measured by overlap in the VisA properties). In Figure 3, we also plot a similar correlation curve for an emerged language obtained by producing messages with randomly initialized and untrained speaker/listener architectures. This emerged language is at random in terms of communicative success; however, the generated messages do show signs of structure, since similar objects obtain somewhat similar messages. This seems to suggest that structured and disentangled pre-linguistic representations are, perhaps, a sufficient condition for the emergence of structured language, especially in neural network-based agents which, due to the nature of representation and information flow, favor similar inputs to trigger similar outputs.

## 4 STUDY 2: REFERENTIAL GAME WITH RAW PIXEL DATA

In this section, we present experiments in which agents receive as input entangled data in the form of raw pixel input, and have to learn to perform visual conceptual processing guided by the communication-based reward.

We use a synthetic dataset of scenes consisting of geometric objects generated using the MuJoCo physics engine (Todorov et al., 2012). We generate RGB images of resolution $124 \times 124$ depicting single object scenes. For each object, we pick one of eight colors (*blue, red, white, black, yellow, green, cyan, magenta*) and five shapes (*box, sphere, cylinder, capsule, ellipsoid*) resulting in 40 combinations, for each of which we generate 100 variations, varying the floor color and the object location in the image. Moreover, we introduce different variants of the game: game **A** with 19 distractors; game **B** with 1 distractor; game **C** with 1 distractor, and with speaker and listener having different viewpoints of the target object (the target object on the listener's side is in a different location); game **D** with 1 distractor, with speaker and listener having different viewpoints, and with balanced numbers of shapes and color (obtained by downsampling from 8 colors to 5 and removing any image containing objects of the 3 disregarded objects). For each game, we create train and test splits with proportions 75/25 (i.e., 3000/1000 for games **A** and **B**, and 1850/650 for games **C** and **D**).

Pre-linguistic objects are presented in the form of pixel input, $o \in [0, 255]^{3 \times 124 \times 124}$. Speaker and listener convert the images $o$ to dense representations $u$, each of them using an 8-layer convolutional neural network (ConvNet). Crucially, we do not pre-train the ConvNets on an object classification task; the only learning signal is the communication-based reward. Despite this fact, we observe that the lower layers of the ConvNets are encoding similar information to a ConvNet pre-trained on ImageNet (Deng et al., 2009).[3] Conceptually, we can think of the whole speaker/listener architecture as an encoder-decoder with a discrete bottleneck (the message). Given our initial positive findings, this reward-based learning signal induced from the communication game setup could be used for class-agnostic large-scale ConvNet training. Moreover, we find that, even though no weights were shared, the agents' conceptual spaces get aligned at different levels, reminiscent of theories of interactive conceptual alignment during dialogue (Garrod & Pickering, 2004) (see Appendix A for the related experiment).

## 4.1 COMMUNICATIVE SUCCESS AND EMERGENT PROTOCOLS

Unlike the experiments of Section 3, where agents start from disentangled representations, starting from raw perceptual input presents a greater challenge: the agents have to establish naming conventions about scenes, while at the same time learning to process the input with their own visual conceptual system. Since we do not pre-train their ConvNets on an object recognition task, the dense representations $u$ used to derive the message contain no bias towards any image- or scene-specific information (e.g, object color, shape or location). The extraction of visual properties is thus driven entirely by the communication game. This contrasts with the cases of Havrylov & Titov (2017) and Lazaridou et al. (2017) who use pre-trained visual vectors, and qualitatively observe that the induced communication protocols encode information about objects. Table 3 presents the results in terms of communicative **train** and **test** success (see Appendix C for additional experiments when having access to gold object attribute classifiers). Moreover, we also report the topographic similarity (column **topographic** $\rho$) between the symbolic attribute-based representations of scenes (floor color, object color, shape and location) and the generated messages.

Overall, despite the challenges posed in this setup due to the raw nature of the data, performance across all games is well above chance, indicating that reinforcement learning agents trained end-to-end are able to establish a communication protocol in this grounded environment. In game **A**, the agents reach 93.7% accuracy, with their lexicon consisting of 1068 messages, describing 3000 training objects. Most importantly, as captured by the positive topographic similarity, agents produce messages that respect (even to a limited degree) the compositional nature of scenes (i.e., objects as bags-of-attributes), indicating that similar scenes receive similar messages. Indeed, by examining their protocol (see Table 4), we find that messages encode in a structurally consistent way information about absolute location of objects, with the message prefix and suffix denoting the horizontal and vertical co-ordinate, respectively. Interestingly, this communication strategy is also typically followed by human players of referential games (Kazemzadeh et al., 2014).

---

[3]Specifically, for all 4000 images we compute two sets of activations, one derived from the speaker's ConvNet and one from a pre-trained ResNet model (He et al., 2016). We then compute all pairwise cosines in the speaker's ConvNet and ResNet space and correlate these values. We find Spearman $\rho$ to be in the range 0.6-0.7 between the first 3 layers of the speaker's ConvNet and the ResNet.

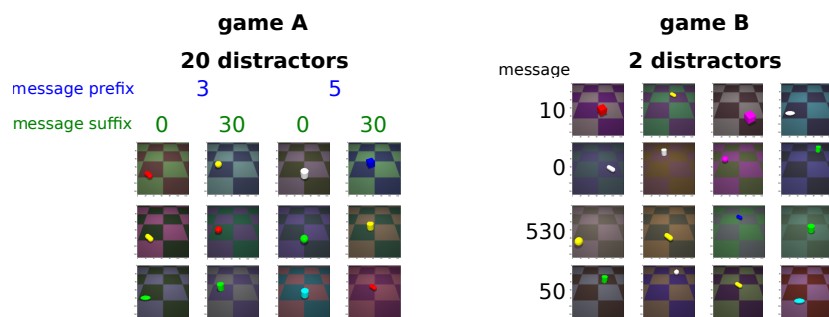

Figure 4: Target images and their associated messages from game **A** and game **B**.

| game | distractors | balanced | viewpoints | lexicon size | random | train | test | topographic $\rho$ |
|------|-------------|----------|------------|--------------|--------|-------|------|--------------------|
| **A** | 20 | No | No | 1068 | 5.0 | 93.7 | 93.6 | 0.13 |
| **B** | 2 | No | No | 13 | 50.0 | 93.2 | 93.4 | 0.006 |
| **C** | 2 | No | Yes | 8 | 50.0 | 86.0 | 85.7 | 0.07 |
| **D** | 2 | Yes | Yes | 5 | 50.0 | 90.4 | 89.9 | 0.06 |

Table 3: Communicative success of agents playing different games. Columns **random**, **train** and **test** report percentage accuracies. Column **topographic** $\rho$ reports the topographic similarity between the symbolic representation of scenes and the generated messages ($p < 0.01$, permutation test).

However, we find the emerged protocols to be very unstable and too grounded in the specific game situation. Small modifications of the game setup, while having close to no negative impact on the communicative performance, can radically alter the form, semantics and interpretability of the communication protocol. In game **B**, performance remains at the same level (93.2%) as game **A**. However, we observe that the protocol consists of 13 unique messages which do not reflect the objects' attributes (as indicated by the close to zero topographic similarity), thus making the messages harder to interpret (see Figure 4 for randomly sampled examples). When we change the viewpoint of the agents in game **C**, biasing them against communicating about absolute object location, the players derive a compact communication protocol consisting of 8 unique messages that describe primarily color. Finally, when color and shape are balanced, as in game **D**, we still observe a bias towards describing the color of objects, with the five induced messages providing a perfect clustering of the objects according to their colors.[4]

In an entangled world, agents do not possess *a priori* visual biases and knowledge of concepts. Since objects can be conceptualized in indefinitely many ways, the type of information encoded in the messages is tied to the environmental pressures; communication behaviour is a function of the environment, which also dictates what data structures can emerge. The implication of this observation is that protocols essentially overfit to the particular game situation, to the degree that they become specialized ad-hoc naming conventions.

Interestingly, the emergence of ad-hoc naming conventions has also been observed during human-human interaction: when participants engage in some specific game situation (e.g., communicating about abstract tangram shapes), they tend to form highly specialized naming conceptions (*conceptual pacts*) that allow them to communicate with maximum efficiency (Brennan & Clark, 1996). While in this study we do not address the issue of how a stable and general language could emerge in entangled worlds, we believe that to alleviate the formation of such ad-hoc communication protocols, it is essential to increase the complexity of the games as well as requiring transfer across a variety of games.

---

[4]Overall, results with REINFORCE in these non-stationary multi-agent environments (where speakers and listeners are learning at the same time from raw pixels) show instability, and (as expected) some of the experimental runs did not converge. However, we observed that the stability of the nature of the protocol (rather than its existence) is mostly influenced by the configuration of the game itself, i.e., how constrained the message space is.

| game | object position | object shape | object color | floor color |
|------|-----------------|--------------|--------------|-------------|
| (random baseline) | (20.0) | (20.0) | (12.0) | (33.0) |
| A | 95.3 | 90.2 | 24.7 | 36.4 |
| B | 88.6 | 41.2 | 63.8 | 45.4 |
| C | 85.9 | 43.5 | 65.8 | 43.8 |
| D | 89.4 | 47.1 | 82.0 | 42.3 |

Table 4: Accuracy of probe linear classifiers of speaker's induced visual representations (all accuracies are in percentage format).

## 4.2 PROBE MODELS

In order to investigate what information gets captured by the speaker's ConvNet, we probe the inferred visual representations $u$ used to derive the message. Specifically, we design 4 probe classifiers for the **color** and **shape** of the object; **object position** which is derived by discretizing each co-ordinate into 3 bins; and **floor color** which is obtained by clustering the RGB color representation of the floor. For each probe, we performed 5-fold cross validation with a linear classifier, and report accuracy results in Table 4. Overall, different games result in visual representations with different predictive power; **object position** is almost always encoded in the speaker's visual representation, even in situations where location of the object is not a good strategy for communication. On the other hand, **object shape** seems to provide less salient information, despite the fact that it is relevant for communication, at least in the **C&D** games.

As expected, the structure and semantics of the emergent protocols are a function of the information captured in the visual representations. The degree to which the agents are able to pull apart the objects' factors of variation impacts their ability to communicate about those factors, with the most extreme case being game **D**, where the message ignores the shape entirely. Thus, disentanglement seems to be a necessary condition for communication, at least in the case of pixel input.

## 5 CONCLUSION

We presented a series of studies investigating the properties of protocols emerging when reinforcement learning agents are trained end-to-end on referential communication games. We found that when agents are presented with disentangled input data in the form of attribute vectors, this inherent compositional structure is successfully retained in the output. Moreover, we showed that communication can also be achieved in cases where agents are presented with raw pixel data, a type of input that aligns better with the raw sensorimotor data that humans are exposed to. At the same time, we found that their ability to form compositional protocols in these cases is hampered by their ability to pull apart the objects' factors of variations. Altogether, we were able to successfully scale up traditional research from the language evolution literature on emergent communication tasks to the contemporary deep learning framework, thus opening avenues to more realistic, and large scale, computational simulations of language emergence with complex image stimuli.

## ACKNOWLEDGEMENTS

We would like to thank Murray Shanahan, Laura Rimell and Gabor Melis for their very helpful feedback on this paper, as well as the rest of the DeepMind language team for many discussions. AL would also like to thank Marco Baroni and Alex Peysakhovich for the email correspondence and discussions from a year ago, which provided inspiration for some of the experiments on this paper.

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

## A  Conceptual alignment of speaker and listener

During conversation, communication allows interlocutors to achieve interactive conceptual alignment (Garrod & Pickering, 2004). We are able to communicate because we have established a common ground and our representations at different levels become aligned (e.g., participants mutually understand that "he" in the conversation refers to Bob). We investigated whether the agents' conceptual systems achieve a similar structural alignment. We measure the alignment in terms of Spearman $\rho$ correlation of the intra-agent pairwise object cosine similarities as calculated via representing objects as activations from ConvNet layers.

Interestingly, we observe a gradual increase in the structural similarity as we represent the objects with layer activations closer to the pixel space. Conceptual spaces are more aligned the closer they are to the raw pixel input ($\rho = 0.97$-$0.91$, depending on the game) and become more dissimilar as the representations become more abstract. We can draw the analogy to language processing, as first ConvNet layers perform some low-level processing analogous to phoneme recognition or word segmentation (and are thus more objective) while higher layers perform more abstract processing, vaguely analogous to semantics and pragmatics (thus, represent more subjective knowledge). In cases of successful communication, speakers' and listeners' conceptual spaces closer to the communication point are structurally very similar ($\rho = 0.85$-$0.62$, depending on the game), however this similarity drops dramatically in cases of failure of communication ($\rho = 0.15$).

## B  Hyperparameter details

All LSTM hidden states of the "speaking" and "listening" module as well and the "seeing" pre-linguistic feed-forward encoders (see Section 3), have dimension 50. The "seeing" pre-linguistic ConvNet encoders (see Section 4) has 8 layers, 32 filters with the kernel size 3 for every layer and with strides $[2, 1, 1, 2, 1, 2, 1, 2]$ for each layer. We use ReLU as activation function as well as batch normalization for every layer.

For learning, we used the Rmsprop optimizer, with learning rate 0.0001. We use a separate value of entropy regularization for each policy. For $\pi^S$ we use 0.01 and for $\pi^L$ we use 0.001. We use a mini-batch of 32.

## C  Communicative success using gold attribute classifiers

We assume a model which has access to perfect attribute classifiers for **color**, **shape** and **object position**, for the latter using a classifier operating on the discretized annotations we obtained in Section 4.2 after quantazing the real-valued object location. For computing the performance of this model using gold attribute classifiers, we first remove from the distractors any candidate not matching the target's attributes and them pick at random. We repeat this experiment for single attribute classifiers and their pairwise combinations. Table 5 reports the communicative success results obtained empirically by averaging across 1000 simulations, alongside the training and test accuracies of the trained agents of Section 4.1 for comparison.

| game | train | test | color | shape | position | color & shape | position & shape | position & color |
|------|-------|------|-------|-------|----------|---------------|------------------|------------------|
| **A** | 93.7 | 93.6 | 37.2 | 24.8 | 69.3 | 80.4 | 92.1 | 95.6 |
| **B** | 93.2 | 93.4 | 93.2 | 90.1 | 97.2 | 98.8 | 99.3 | 99.4 |
| **C** | 86.0 | 85.7 | 93.2 | 90.1 | - | 98.8 | - | - |
| **D** | 90.4 | 89.9 | 89.6 | 89.2 | - | 98.5 | - | - |

Table 5: Communicative success of trained models from Section 4.1 (**train** and **test**) as well as models with access to gold classifiers. All accuracies are in percentage format.

