# OpenReview forum: "Emergence of Linguistic Communication from  Referential Games with Symbolic and Pixel Input"
_ICLR.cc/2018/Conference — Accept (Oral)_

### Official Review · AnonReviewer1 · 2017-11-12
**Very interesting paper, writeup could be clearer**

**Rating:** 9
**Confidence:** 5

**Review:**

This paper presents a set of studies on emergent communication protocols in referential games that use either symbolic object representations or pixel-level representations of generated images as input. The work is extremely creative and packed with interesting experiments.

I have three main comments.

* CLARITY OF EXPOSITION

The paper was rather hard to read. I'll provide some suggestions for improvement in the minor-comments section below, but one thing that could help a lot is to establish terminology at the beginning, and be consistent with it throughout the paper: what is a word, a message, a protocol, a vocabulary, a lexicon? etc.

* RELATION BETWEEN VOCABULARY SIZE AND PROTOCOL SIZE

In the compositional setup considered by the authors, agents can choose how many basic symbols to use and the length of the "words" they will form with these symbols. There is virtually no discussion of this interesting interplay in the paper. Also, there is no information about the length distribution of words (in basic symbols), and no discussion of whether the latter was meaningful in any way.

* RELATION BETWEEN CONCEPT-PROPERTY AND RAW-PIXEL STUDIES

The two studies rely on different analyses, and it is difficult to compare them. I realize that it would be impossible to report perfectly comparable analyses, but the authors could at least apply the "topographic" analysis of compositionality in the raw-pixel study as well, either by correlating the CNN-based representational similarities of the Speaker with its message similarities, or computing similarity of the inputs in discretized, symbolic terms (or both?).

* MINOR/DETAILED COMMENTS

Section 1

How do you think emergent communication experiments can shed light on language acquisition?

Section 2

In figure 1, the two agents point at nothing.

\mathbf{v} is a set, but it's denoted as a vector. Right below that, h^S is probably h^L?

all candidates c \in C: or rather their representations \mathbf{v}?

Give intuition for the reward function.

Section 3

We use the dataset of Visual Attributes...: drop "dataset"

I think the pre-linguistic objects are not represented by 1-hot, but binary vectors.

do care some inherent structure: carry

Note that symbols in V have no pre-defined semantics...: This is repeated multiple times.

Section 3

I couldn't find simulation details: how many training elements, and how is training accuracy computed? Also, "training data", "training accuracy" are probably misleading terms, as I suppose you measured performance on new combinations of objects.

I find "Protocol Size" to be a rather counterintuitive term: maybe call Vocabulary Size "Alphabet Size", and Protocol Size "Lexicon Size"?

State in Table 1 caption that the topographic measure will be explained in a later section. Also, the -1 is confusing: you can briefly mention when you introduce the measure that since you correlate a distance with a similarity you expect an inverse relation? Also, you mention in the caption that all Spearman rhos are significant, but where are they presented again?

Section 3.2

Does the paragraph starting with "Note that the distractor" refer to a figure or table that is not there? If not, it should be there, since it's not clear what are the data that support your claims there. Also, you should explain what the degenerate strategy the agents find is.

Next paragraph:

- I find the usage of "obtaining" to refer to the relation between messages and objects strange.

- in which space are the reported pairwise similarities computed?

- make clear that in the non-uniform case confusability is less influenced by similarity since the agents must learn to distinguish between similar objects that naturally co-occur (sheep and goats)

- what is the expected effect on the naturalness of the emerged language?

Section 3.3

adhere to, the ability to: "such as" missing?

Is the unigram chimera distribution inferred from the statistics over the distribution of properties across all concepts or what? (please clarify.)

In Tables 2 and 3, why is vocabulary size missing?

In Table 2, say that the protocol size columns report novel message percentage **for the "test" conditions***

Figure 2: spelling of Levensthein

Section 3.3.2

while for languages (c,d)... something missing.

with a randomly initialized...: no a

More importantly, I don't understand this "random" setup: if architecture was fixed and randomly initialized, how could something be learned about the structure of the data?

Section 4

Refer to the images the agents must communicate about as "scenes", since objects are just a component of them.

What are the absolute sizes of train and test splits?

Section 4.1

we do not address this issue: the issue

Section 4.2

at least in the game C&D: games

Why is Appendix A containing information that logically follows that in Appendix B?

---

> ### Author Response · Authors · 2018-01-03
> **Response to AnonReviewer1**
>
> We would like to thank the reviewer for their review. We found their comments extremely helpful and we are in the process of updating the manuscript accordingly. We will upload the revised paper tomorrow. In the meantime, we respond here to the major comments.
>
>
> <review>
> * CLARITY OF EXPOSITION
> </review>
> We will introduce the terminology together with the description of the game.
>
> <review>
> * RELATION BETWEEN VOCABULARY SIZE AND PROTOCOL SIZE
> </review>
> Without any explicit penalty on the length of the messages (Section 2), agents are not motivated to produce shorter messages (despite the fact that as the reviewer points, agents can decide to do so) since this constrains the space of messages (and thus the possibility of the speaker and listener agreeing on a successful naming convention), opting thus to always make use of the maximum possible length. When we introduced a penalty on the length of the message (Section 3), agents produced shorter messages for the ambiguous messages since this strategy maximizes the total expected reward.
>
>
> <review>
> * RELATION BETWEEN CONCEPT-PROPERTY AND RAW-PIXEL STUDIES
> </review>
> Thanks for the suggestion. Correlating CNN-based representations with message similarities would not yield any new insight since these representations are the input to the message generation process. However, we ran the analysis on the symbolic representations of the images (location cluster, color, shape, floor color cluster) and the messages and found that the topographic similarities of the games are ordered as follows (in parentheses we report the topographic $\rho$): game A (0.13) > game C (0.07) > game D (0.06) > game B (0.006).
> This ordering is in line with our qualitative analysis of the protocols presented in Section 4.1.
>
> <review>
> Figures/Tables for "Note that the distractor"paragraph and degenerate strategy.
> </review>
> We will include in the manuscript the training curves that this paragraph refers to.
> The degenerate strategy is that of picking a target at random from the topically relevant set of distractors, thus reducing the effective size of distractors.
>
> <review>
> "random" setup...
> </review>
> Despite the fact that the weights of the networks are random, since the message generation is a parametric process, similar inputs will tend to generate similar outputs, thus producing messages that retain (at least to some small degree) the structure of the input data, despite the fact that there is no learning at all.

---

> > ### Comment · AnonReviewer1 · 2018-01-04
> > **Thanks for the clarifications**
> >
> > Thanks for the clarifications, and looking forward to the revised paper.

---

### Official Review · AnonReviewer2 · 2017-11-26

**Rating:** 7
**Confidence:** 4

**Review:**

--------------
Summary:
--------------
This paper presents a series of experiments on language emergence through referential games between two agents. They ground these experiments in both fully-specified symbolic worlds and through raw, entangled, visual observations of simple synthetic scenes. They provide rich analysis of the emergent languages the agents produce under different experimental conditions. This analysis (especially on raw pixel images) make up the primary contribution of this work.


--------------
Evaluation:
--------------
Overall I think the paper makes some interesting contributions with respect to the line of recent 'language emergence' papers. The authors provide novel analysis of the learned languages and perceptual system across a number of environmental settings, coming to the (perhaps uncontroversial) finding that varying the environment and restrictions on language result in variations in the learned communication protocols.

In the context of existing literature, the novelty of this work is somewhat limited -- consisting primarily of the extension of multi-agent reference games to raw-pixel inputs. While this is a non-trivial extension, other works have demonstrated language learning in similar referring-expression contexts (essentially modeling only the listener model [Hermann et.al 2017]).

I have a number of requests for clarification in the weaknesses section which I think would improve my understanding of this work and result in a stronger submission if included by the authors.

--------------
Strengths:
--------------
- Clear writing and document structure.


- Extensive experimental setting tweaks which ablate the information and regularity available to the agents. The discussion of the resulting languages is appropriate and provides some interesting insights.


- A number of novel analyses are presented to evaluate the learned languages and perceptual systems.


--------------
Weaknesses:
--------------
- How stable are the reported trends / languages across multiple runs within the same experimental setting? The variance of REINFORCE policy gradients (especially without a baseline) plus the general stochasticity of SGD on randomly initialized networks leads me to believe that multiple training runs of these agents might result is significantly different codes / performance. I am interested in hearing the author's experiences in this regard and if multiple runs present similar quantitative and qualitative results. I admit that expecting identical codes is unrealistic, but the form of the codes (i.e. primarily encoding position) might be consistent even if the individual mappings are not).


- I don't recall seeing descriptions of the inference-time procedure used to evaluate training / test accuracy. I will assume argmax decoding for both speaker and listener. Please clarify or let me know if I missed something.


- There is ambiguity in how the "protocol size" metric is computed. In Table 1, it is defined as 'the effective number of unique message used'. This comes back to my question about decoding I suppose, but does this count the 'inference-time' messages or those produced during training?
Furthermore, Table 2 redefines "protocol size" as the percentage of novel message. I assume this is an editing error given the values presented and take these columns as counts. It also seems "protocol size" is replaced with the term "lexicon" from 4.1 onward.

- I'm surprised by how well the agents generalize in the raw pixel data experiments. In fact, it seems that across all games the test accuracy remains very close to the train accuracy.

Given the dataset is created by taking all combinations of color / shape and then sampling 100 location / floor color variations, it is unlikely that a shape / color combo has not been seen in training. Such that the only novel variations are likely location and floor color. However, taking Game A as an example, the probe classifiers are relatively poor at these attributes -- indicating the speaker's representation is not capturing these attributes well. Then how do the agents effectively differentiate so well between 20 images leveraging primarily color and shape?

I think some additional analysis of this setting might shed some light on this issue. One thought is to compute upper-bounds based on ground truth attributes. Consider a model which knows shape perfectly, but cannot predict other attributes beyond chance. To compute the performance of such a model, you could take the candidate set, remove any instances not matching the ground truth shape, and then pick randomly from the remaining instances. Something similar could be repeated for all attributes independently as well as their combinations -- obviously culminating in 100% accuracy given all 4. It could be that by dataset construction, object location and shape are sufficient to achieve high accuracy because the odds of seeing the same shape at the same location (but different color) is very low.

Given these are operations on annotations and don't require time-consuming model training, I hope to see this analysis in the rebuttal to put the results into appropriate context.


- What is random chance for the position and floor color probe classifiers? I don't think it is mentioned how many locations / floor colors are used in generation.


- Relatively minor complaint: Both agents are trained via the REINFORCE policy gradient update rule; however, the listener agent makes a fairly standard classification decision and could be trained with a standard cross-entropy loss. That is to say, the listener policy need not make intermediate discrete policy decisions. This decision to withhold available supervision is not discussed in the paper (as far as I noticed), could the authors speak to this point?



--------------
Curiosities:
--------------
- I got the impression from the results (specifically the lack of discussion about message length) that in these experiments agents always issued full length messages even though they did not need to do so. If true, could the authors give some intuition as to why? If untrue, what sort of distribution of lengths do you observe?

- There is no long term planning involved in this problem, so why use reinforcement learning over some sort of differentiable sampler? With some re-parameterization (i.e. Gumbel-Softmax), this model could be end-to-end differentiable.


--------------
Minor errors:
--------------
[2.2 paragraph 1] LSTM citation should not be in inline form.
[3 paragraph 1] 'Note that these representations do care some' -> carry
[3.3.1 last paragraph] 'still able comprehend' --> to


-------
Edit
-------
Updating rating from 6 to 7.

---

> ### Author Response · Authors · 2018-01-01
> **Response to AnonReviewer2 (part 1)**
>
> We thank the reviewer for their thorough review. We respond to the comments raised while we are in the process of making the necessary changes in the manuscript.
>
> <review>
> How stable are results?
> </review>
> Overall, results with REINFORCE in these non-stationary multi-agent environments (where speakers and listeners are learning at the same time) show instability, and -- as expected -- some of the experimental runs did not converge. However, we believe that the stability of the nature of the protocol (rather than its existence) is mostly influenced by the configuration of the game itself, i.e., how constrained the message space is. As an example, games C & D impose constraints on the nature of the protocol since location encoding location on the messages is not an acceptable solution -- on runs that we had convergence, the protocols would always communicate about color. The same holds for game A (position is a very good strategy since it uniquely identifies objects combined with the environmental pressure of many distractors). However, game B is more unconstrained in nature and the converged protocols were more varied. We will include a discussion of these observations in the updated manuscript.
>
> <review>
> Inference time procedure
> </review>
> The reviewer is correct. At training time we sample, at test time we argmax. We will clarify this.
>
> <review>
> Protocol size vs lexicon
> </review>
> Thank you for pointing this out. We will clarify the terminology.
> Protocol size (or lexicon -- we will remove this term and use protocol size only) is the number of invented messages (sequences of symbols).
> In Table 1, we report the protocol size obtained with argmax on training data.
> In Table 2, we report the number of novel messages, i.e., messages that were not generated for the training data, on 100 novel objects.
>
> <review>
> Generalization on raw pixel data -- training and test accuracy are close
> </review>
> This observation is correct. By randomly creating train and test splits, chances are that the test data contain objects of seen color and shape combination but unseen location. Neural networks (and any other parametric model) do better in these type of “smooth” generalizations caused by a continuous property like location.
>
> <review>
> However, taking Game A as an example, the probe classifiers are relatively poor at these attributes -- indicating the speaker's representation is not capturing these attributes well.
> Then how do the agents effectively differentiate so well between 20 images leveraging primarily color and shape?
> </review>
> In Game A, agents differentiate 20 objects leveraging primarily object position rather than color and shape.
> In Game A, the listener needs to differentiate between 20 objects, and so, communicating about color and shape is not a good strategy as there are chances that there will be some other red cube, for example, on the distractor list. The probe classifiers are performing relatively poorly on these attributes (especially on the object color) whereas they perform very well on position (which is in fact a good strategy), which as we find by our analysis is what the protocol describes. We note that location is a continuous variable (which we discretize only for performing the probe analysis in Section 4.2) and so it is very unlikely that two objects have the same location, thus uniquely identifying objects among distractors. This is not the case for games C & D since the listener sees a variation of the speaker’s target.
> Moreover, we note, that object location is encoded across all games.
>
> <review>
> Upper-bound analysis based on ground truth attributes.
> </review>
> We agree with the reviewer that an upper-bound analysis relying on gold information of objects will facilitate the exposition of results. Note that since location is a continuous variable, ground truth of location is not relevant.
> 		color 	shape  color & shape
> A		0.37	         0.24	0.80
> B & C 	0.93	         0.90	0.98
> D		0.89	         0.89	0.98
>
> We could perform the same analysis by discretizing the location in the same way we performed the probe analysis in Section 4.2, however, the upper-bound results depend on the number of discrete locations we derive.
> 		location	    color & location	shape & location
> A		0.69		         0.95			0.92
> B 		0.97		         0.99			0.99
> (for C and D results for location are not applicable)

---

> ### Author Response · Authors · 2018-01-01
> **Response to AnonReviewer2 (part 2)**
>
>
> <review>
> random chance of probe classifiers.
> </review>
> When generating the dataset, we sample locations and floor colors from a continuous scale. For the probe classifiers, we quantize location by clustering each coordinate in 5 clusters (and thus accuracy is reported by averaging the performance of the x and y probe classifiers with chance being at 20% for each co-ordinate) and floor colors in 3 clusters (with chance being at 33%). We will include the chance levels in Table 4.
>
> <review>
> Why not use cross-entropy loss for listener?
> </review>
> We decided to train both agents via REINFORCE for symmetry. Given the nature of the listener’s choice, we don’t anticipate full supervision to have an effect other than speeding up learning.
>
>
> <review>
> What about message length?
> </review>
> Without any explicit penalty on the length of the messages (Section 2), agents are not motivated to produce shorter messages (despite the fact that as the reviewer points, agents can decide to do so) since this constrains the space of messages (and thus the possibility of the speaker and listener agreeing on a successful naming convention). When we introduced a penalty on the length of the message (Section 3), agents produced shorter messages for the ambiguous messages (since this strategy maximizes the total expected reward).
>
> <review>
> Why use reinforcement learning over some sort of differentiable sampler?
> </review>
> While a differentiable communication channel would make learning faster, it goes against the basic and fundamental principles of human communication (and also against how this phenomenon is studied in language evolution).  Simply put, having a differentiable channel would mean in practice that speakers can back-propagate through listeners’ brains (which unfortunately is not the case in real life :)) We wanted to stay as close as possible to this communication paradigm, thus using a discrete communication channel.

---

### Official Review · AnonReviewer3 · 2017-11-28
**Explores interesting issues,  but needs more quantitative analysis and has limited novelty**

**Rating:** 5
**Confidence:** 4

**Review:**

This paper presents an analysis of the communication systems that arose when neural network based agents played simple referential games. The set up is that a speaker and a listener engage in a game where both can see a set of possible referents (either represented symbolically in terms of features, or represented as simple images) and the speaker produces a message consisting of a sequence of numbers while the listener has to make the choice of which referent the speaker intends. This is a set up that has been used in a large amount of previous work, and the authors summarize some of this work. The main novelty in this paper is the choice of models to be used by speaker and listener, which are based on LSTMs and convolutional neural networks. The results show that the agents generate effective communication systems, and some analysis is given of the extent to which these communications systems develop compositional properties – a question that is currently being explored in the literature on language creation.

This is an interesting question, and it is nice to see worker playing modern neural network models to his question and exploring the properties of the solutions of the phone. However, there are also a number of issues with the work.

1. One of the key question is the extent to which the constructed communication systems demonstrate compositionality. The authors note that there is not a good quantitative measure of this. However, this is been the topic of much research of the literature and language evolution. This work has resulted in some measures that could be applied here, see for example Carr et al. (2016): http://www.research.ed.ac.uk/portal/files/25091325/Carr_et_al_2016_Cognitive_Science.pdf

2. In general the results occurred be more quantitative. In section 3.3.2 it would be nice to see statistical tests used to evaluate the claims. Minimally I think it is necessary to calculate a null distribution for the statistics that are reported.

3. As noted above the main novelty of this work is the use of contemporary network models. One of the advantages of this is that it makes it possible to work with more complex data stimuli, such as images. However, unfortunately the image example that is used is still very artificial being based on a small set of synthetically generated images.

Overall, I see this as an interesting piece of work that may be of interest to researchers exploring questions around language creation and language evolution, but I think the results require more careful analysis and the novelty is relatively limited, at least in the way that the results are presented here.

---

> ### Author Response · Authors · 2017-12-18
> **Response to AnonReviewer3**
>
> We thank the reviewer for their comments.
> For replying, we copy-paste the relevant part and comment on it.
>
> <review> 1. One of the key question ...  Carr et al. (2016): http://www.research.ed.ac.uk/portal/files/25091325/Carr_et_al_2016_Cognitive_Science.pdf"
> </review>
>
> We agree with the reviewer that there are good existing measures. Our point was only that there is no mathematical definition and hence no definitive measure. In fact, we do include such a measure found in the literature on language evolution. Our topographic similarity measure (which is introduced by Brighton & Kirby (2006)) is in line with the measure introduced in 2.2.3 in Carr et al.. In Carr et al, the authors correlate Levenshtein message distances and triangle dissimilarities (as obtained from humans). In our study, we correlate Levenshtein message distances and object dissimilarities as obtained by measuring cosine distance of the object feature norms (which are produced by humans). We will make sure to make this connection to previous literature explicit in our description of the measure.
>
> <review>
> 2. In general the results occurred be more quantitative....statistics that are reported.
> </review>
>
> We agree with the reviewer that statistical tests are important, and we politely point out that our claims on 3.3.2 are in fact based on the reported numbers in Table 1 “topographic ρ” column.  However, we will evaluate the statistical significance of the “topographic ρ” measure by calculating the null distribution via a repeated shuffling of the Levenshtein distances (or an additional test if the reviewer has an alternative suggestion).
>
> <review>
> 3. As noted above the main novelty of this work is the use of contemporary network models
> </review>
>
> We believe the novelty of this work is to take the well-defined and interesting questions that the language evolution literature has posed and try to scale them up to contemporary deep learning models and materials, i.e., realistic stimuli in terms of objects and their properties (see Section 3), raw pixel stimuli (see Section 4) and neural network architectures (see Section 2). This kind of interdisciplinary work can not only inform current models on their strengths and weaknesses (as we note in Section 4 we find that neural networks starting from raw pixels cannot out-of-the-box process easily stimuli in a compositional way), but also open up new possibilities for language evolution research in terms of more realistic model simulations. We believe that this might not have been clear from the manuscript and will update the abstract and conclusion to reflect the premises of the work.
>
> <review>
> One of the advantages of this is that it makes it possible to work with more complex data stimuli, such as images. However, unfortunately the image example that is used is still very artificial being based on a small set of synthetically generated images.
> </review>
>
> More complex image stimuli and realistic simulations is where we are heading. However, we (as a community) first need to understand how these models behave with raw pixels before scaling them up to complex stimuli. The nature of this work was to lay the groundwork on this question and investigate the properties of protocols in controlled (yet realistic in terms of nature) environments where we can tease apart clearly the behaviour of the model given the small number of variations of the pixel stimuli (object color/shape/position and floor color). Performing the type of careful analysis we did for complex scenes is substantially harder due to the very large number of factors we would have to control (diverse objects of multiple colors, shapes, sizes, diverse backgrounds etc) so it puts into question to what degree we could have achieved a similar degree of introspection by immediately using more complex datasets in the current study.
>
> <review>
> Overall, I see this as an interesting piece of work that may be of interest to researchers exploring questions around language creation and language evolution, but I think the results require more careful analysis and the novelty is relatively limited, at least in the way that the results are presented here.
> </review>
>
> We will upload an updated version of our paper by the end of this week containing
> 1) the statistical test of the null distribution
> 2) clarifications regarding the topographic measure and
> 3) we will clarify the main contributions of this work and better relate it to the existing literature in language evolution
>
> Moreover, we would be really happy to conduct further analyses and clarify the exposition of results. If the reviewer has specific suggestions on this, we would like to hear them in order to improve the quality of the manuscript and strengthen our submission.

---

### Author Response · Authors · 2018-01-04
**Revised manuscript**

We would like to thank all the reviewers for their thoughtful and detailed feedback. We particularly thank them for recognizing that this is an interesting piece of work.

We have now revised our manuscript to address the concerns raised by the reviewers, hopefully producing a stronger and clearer submission. The most significant changes are:

	(as asked by AnonReviewer3)
* We have added statistical tests (permutation test) to support claims regarding the results of the topographic similarity
* We have added 2 sentences in the abstract and conclusion to make clear our contributions on extending work in the language evolution literature to contemporary DL materials.

	(as asked by AnonReviewer2)
* We have added in the Appendix C a new experiment on  communicative success  on the raw pixel data with models operating  on gold attribute classifiers
* We have added a comment about instability of REINFORCE affecting nature of protocols in same of the experimental setups of Section 4

	(as asked by AnonReviewer1)
* We made all the requested clarifications (thanks again for the detailed review)
* Added Figure 2 to visually illustrate the claims in Section 3.2
* Added topographic similarity measurements for Section 4 (Table 3) which strengthen the findings of the qualitative analysis of game A producing structurally consistent messages.

---

### Decision · Program_Chairs · 2018-01-29
**ICLR 2018 Conference Acceptance Decision**

**Decision:**

Accept (Oral)

**Comment:**

Important problem (analyzing the properties of emergent languages in multi-agent reference games), a number of interesting analyses (both with symbolic and pixel inputs), reaching a finding that varying the environment and restrictions on language result in variations in the learned communication protocols (which in hindsight is that not surprising, but that's hindsight). While the pixel experiments are not done with real images, it's an interesting addition the literature nonetheless.